# The Endophytic Microbiome of Wild Grapevines *Vitis amurensis* Rupr. and *Vitis coignetiae* Pulliat Growing in the Russian Far East

**DOI:** 10.3390/plants12162952

**Published:** 2023-08-15

**Authors:** Olga A. Aleynova, Nikolay N. Nityagovsky, Alexey A. Ananev, Andrey R. Suprun, Zlata V. Ogneva, Alina A. Dneprovskaya, Alina A. Beresh, Alexey P. Tyunin, Alexandra S. Dubrovina, Konstantin V. Kiselev

**Affiliations:** 1Laboratory of Biotechnology, Federal Scientific Center of the East 27Asia Terrestrial Biodiversity, Far Eastern Branch of the Russian Academy of Sciences, 690022 Vladivostok, Russia; 2The School of Natural Sciences, Far Eastern Federal University, 690090 Vladivostok, Russia

**Keywords:** biodiversity, endophytes, fungi, bacteria, microbiome, wild relatives of domesticated plants, *16S*, *ITS1*, next-generation sequencing, *Vitis amurensis*, *Vitis coignetiae*

## Abstract

Many grape endophytic microorganisms exhibit high potential for suppressing the development of grape diseases and stimulating grapevine growth and fitness, as well as beneficial properties of the crop. The microbiome of wild grapevines is a promising source of biocontrol agents, which can be beneficial for domesticated grapevines. Using next-generation sequencing (NGS) and classical microbiology techniques, we performed an analysis of bacterial and fungal endophytic communities of wild grapevines *Vitis amurensis* Rupr. and *Vitis coignetiae* Pulliat growing in the Russian Far East. According to the NGS analysis, 24 and 18 bacterial taxa from the class level were present in *V. amurensis* and *V. coignetiae* grapevines, respectively. Gammaproteobacteria (35%) was the predominant class of endophytic bacteria in *V. amurensis* and Alphaproteobacteria (46%) in *V. coignetiae.* Three taxa, namely *Sphingomonas*, *Methylobacterium,* and *Hymenobacter,* were the most common bacterial genera for *V. amurensis* and *V. coignetiae*. Metagenomic analysis showed the presence of 23 and 22 fungi and fungus-like taxa of class level in *V. amurensis* and *V. coignetiae*, respectively. The predominant fungal classes were Dothideomycetes (61–65%) and Tremellomycetes (10–11%), while *Cladosporium* and *Aureobasidium* were the most common fungal genera in *V. amurensis* and *V. coignetiae,* respectively. A comparative analysis of the endophytic communities of *V. amurensis* and *V. coignetiae* with the previously reported endophytic communities of *V. vinifera* revealed that the bacterial biodiversity of *V. amurensis* and *V. coignetiae* was similar in alpha diversity to *V. vinifera*’s bacterial biodiversity. The fungal alpha diversity of *V. amurensis* and *V. coignetiae* was statistically different from that of *V. vinifera*. The beta diversity analysis of bacterial and fungal endophytes showed that samples of *V. vinifera* formed separate clusters, while *V. amurensis* samples formed a separate cluster including *V. coignetiae* samples. The data revealed that the endophytic community of bacteria and fungi from wild *V. amurensis* was richer than that from *V. coignetiae* grapes and cultivated *V. vinifera* grapes. Therefore, the data obtained in this work could be of high value in the search for potentially useful microorganisms for viticulture.

## 1. Introduction

Endophytic microorganisms have been defined as fungi and bacteria (less often as archaea, protists, and viruses) colonizing internal plant tissues [1]. These organisms reside in the living tissues of the host plant and do so in a variety of relationships ranging from symbiotic to pathogenic [2]. Endophytes are involved in the synthesis of siderophores, the production of plant hormones and nitrogen fixation, solubilization of immobilized phosphorus, nutrient cycling, production of volatile organic compounds, and stress tolerance [3]. These endophytes increase plant nutrient uptake and induce plant resistance to pathogens, osmotic stress, heavy metals, xenobiotic contaminants, and other forms of abiotic stress [4]. Endophytic microorganisms may assist in plant pathogen protection, making endophytes useful in the biological control of plant diseases and agricultural pests [5]. Additionally, the microbial diversity of plant endophytic communities may also be important for the postharvest quality and decay of fruits, including grapes [6].

Microbial endophytism exists in virtually every plant on earth, including wild-growing plants. Endophytic microbes, such as *Alternaria* sp., isolated from wild relatives of the domesticated plant families Lamiaceae, Polygonaceae, Rhodomelaceae, Lythraceae, Rubiaceae, Sapindaceae, Vitaceae, Euphorbiaceae, Cupressaceae, Fabaceae, Calophyllaceae, and Cactaceae, were introduced as bio-inoculants in their close cultivars to reduce biotic stress, supporting plant growth successfully in nutrient-deficient soil and providing resistance against aphids [7]. It has been hypothesized that endophytes that are recoverable from crops’ wild relatives would be more compatible with the related crops for both inoculation and colonization purposes. For example, the endophytic *Penicillium* species, isolated from a squill (*Urginea maritima*) leaf, secretes mycophenolic acid, which inhibits the growth of phytopathogenic fungi [8]. It has been shown that the fungal endophyte *Epichloë bromicola,* derived from wild barley (*Hordeum brevisubulatum*), has the potential to increase salt tolerance in cultivated barley under salt stress [9]. The microbial consortium of *Rahnella* sp. (N-fixer), *Bacillus tropicus* (P-solubilizer), and *Bacillus megaterium* (K-solubilizer), isolated from *Aegilops kotschyi*, resulted in the enhancement of wheat crop growth, as well as physiological parameters, including shoot/root length, fresh/dry weight, and the content of chlorophyll, carotenoids, total soluble sugar, phenolics, and flavonoids, compared to uninoculated controls [10]. In addition, endophytes isolated from non-related plants also exhibit plant growth-promoting activities and induce stress tolerance and pathogen resistance. For example, endophytic bacteria isolated from *Mimosa pudica* were able to colonize two very different legumes, *Phaseolus vulgaris* and *M. pudica,* and showed biotechnological potential for plant growth-promoting activity and the production of indoleacetic acid and siderophores [11].

Among the most serious problems faced in vine cultivation are fungal diseases that cause losses in grape production and fruit quality [5]. It is worth pointing out that phytosanitary protection through chemicals may have a negative impact on wine quality, especially through the production of large of amounts tartaric acid, which can be damaging to wine production due to production of salts with low solubility, which can precipitate and thus threaten wine stability [12]. Endophytic microorganisms may be promising alternatives to pesticides for plant protection from fungal diseases [5]. Most studies that have investigated the endophytic communities in grapevines have focused on *Vitis vinifera* cultivars [13,14,15,16,17,18,19,20,21].

Some endophytic bacterial strains of domesticated grapes show bioactive properties. For instance, grapevine-endophytic *Trichoderma* isolates promoted plant growth; in particular, *Trichoderma* treatments significantly increased both the bud burst ratio and bud burst vigor index; grape must parameters, such as the Brix degrees; and the extract, D-glucose, and D-fructose concentrations [22]. Additionally, a new *Trichoderma* strain isolated from the wood of *V. vinifera* cv. Tempranillo has the potential to reduce the colonization of *Phaeoacremonium minimum;* therefore, inoculation with this biological control agent can protect the plant by limiting the development of grapevine trunk diseases, such as esca [23]. Moloinyane and colleagues showed that inoculation of grapevine plants with the endophytic fungus *Beauveria bassiana* induced the synthesis of a markedly higher diversity of anti-insect volatile chemicals in the leaf extracts of fungus-exposed plants compared to the untreated plants [24].

There have also been studies implying the possibility of using endophytic microorganisms as fine-tuning regulators to shape the quality and characteristics of wine grapes [25]. The sources and persistence of wine-relevant microbiota in vineyards critically impact wine’s quality. Site-specific variations in microbiota within and between vineyards may contribute to regional wine characteristics. These variations includes distinctions in microbiomes and microbiota at the strain level, which can contribute to wine flavor and aroma, supporting the role of microbes in the accepted notion of terroir as a biological phenomenon [26].

Endophytes of wild grapevines are a poorly investigated group of microorganisms that represent an abundant and dependable source of bioactive and chemically novel compounds with potential for exploitation in a wide variety of medical, agricultural, and industrial contexts [2]. In support of this assumption, the endophytic strain *Bacillus* K1, isolated from wild grape *Vitis heyneana* Roem. et Schult, showed high antifungal activity against *Botrytis cinerea* both in vitro and in vivo [27]. Additionally, it was found that *Bacillus velezensis* KOF112, which was isolated from the Japanese indigenous wine grape *Vitis* sp. cv. Koshu, inhibited mycelial growth of *B. cinerea* (downy mildew), *Colletotrichum gloeosporioides* (gray mold), and *Phytophthora infestans* (and ripe rot) [28]. An endophytic fungus, *Albifimbria verrucaria*, isolated from the leaves of the Amur grape cultivar Shuangyou, inhibited growth of *B. cinereal* and has potential as a biocontrol agent to control grape gray mold [29]. Therefore, studying endophytic communities of wild grapes is of considerable interest.

The wild-growing Amur grapevine *Vitis amurensis* Rupr. often grows on hillsides or in ravine on hillsides at elevations of 200–2100 m, and it exhibits strong tolerance to cold [30]. This species is a highly resistant to such widespread grapevine diseases as powdery mildew [31], grape white rot, and anthracnose [32]. *Vitis coignetiae* Pulliat contains an abundance of anthocyanins, the anticancer activities of which have been demonstrated in vitro and in vivo regarding apoptosis, cancer invasion, and angiogenesis [33]. Moreover, several previous studies have shown that extracts or compounds of the *V. coignetiae* fruit displayed valuable pharmacological activities, including hepatoprotective [34], anti-inflammatory [35], vasorelaxant [36], antioxidant [37], and anti-cariogenic [38] properties.

The current study aimed to investigate the bacterial and fungal endophytic communities of *V. amurensis* and *V. coignetiae* using a genomic approach (next-generation sequencing or NGS) and classical microbiology techniques (endophyte sowing). This research aimed to isolate and characterize the endophytes from *V. amurensis* and *V. coignetiae,* as well as to find an attractive source of natural products that can be used in agriculture, industry, and medicine.

## 2. Results

### 2.1. High-Throughput Sequencing

Using a genomic approach (next-generation sequencing (NGS)), a total of 12,205,098 *16S* and 4,305,263 *ITS1* paired-end reads were obtained. After paired-end alignments, quality filtering, and deletion of chimeric, mitochondria, chloroplast, *Viridiplantae*, and unidentified sequences, a total of 7,324,582 *16S* and 1,061,426 *ITS1* sequences were generated from 62 plant samples (4–6 samples from each plant) (Appendix A). For the *16S* data, the average and median read numbers of the samples were 118,138 and 64,494 respectively. For the *ITS1* data, the average and median read numbers of the samples were 17,120 and 15,348 respectively.

### 2.2. Biodiversity of Endophytic Bacteria in Vitis amurensis and Vitis coignetiae

Totals of 7,065,631 and 258,951 reads were used for phyla descriptions of endophytic bacteria from *V. amurensis* and *V. coignetiae*, respectively. According to metagenomic analysis of *16S* sequences, 217 and 162 taxa at the genus level with relative representation greater than 0.1% represented the bacterial community in different *V. amurensis* and *V. coignetiae* grapevines, respectively (Figure 1b and Figure 2b). These genera belonged to 24 and 18 taxa at the class level in *V. amurensis* and *V. coignetiae*, respectively (Figure 1a and Figure 2a).

We found that the bacterial diversity of *V. amurensis* and *V. coignetiae* grapevines was influenced by the location of the plant. The beta diversity data showed significant difference between different plants of *V. amurensis and V. coignetiae* performed by PERMANOVA testing (79% variance, *p* < 0.001) (Appendix A).

For *V. amurensis*, the dominant classes in the bacterial communities were Gammaproteobacteria (35%), followed by Alphaproteobacteria (24%), Actinobacteria (17%), Bacteroidia (17%), and Bacilli (5%) (Figure 1a). Interestingly, the highest percentage of the dominant class Gammaproteobacteria was detected in the *V. amurensis* grape samples collected from locations with high anthropogenic impact, namely those near Ivanovka village (64%), near Vladivostok (45% and 64%), in the vineyard “Makarevich” (45%), in Litovko village (47%), and in the greenhouse at the Laboratory of Biotechnology, Federal Scientific Center of the East Asia Terrestrial Biodiversity (35%) (Figure 1a). In grape samples collected at a distance from human settlements, the dominant bacterial class was Alphaproteobacteria (29–45%), in samples from Russky Island, Rikord Island, Sakhalin Island, and the Verkhneussuriysky Research Station (Figure 1a).

The largest number of taxa from the genus level was present in grapevines from locations with a higher influence of human activity, namely Ivanovka (177 genera), near Vladivostok (163 and 162 genera), and the vineyard “Makarevich” (161 genera) (Figure 1b, Appendix A). Among the taxa, 53 bacterial taxa were found in all analyzed *V. amurensis* grapevine samples (Figure 1b, Appendix A). Interestingly, each *V. amurensis* sample location had 1–4 unique genera (Figure 1b).

Using a classical microbiology technique (bacterial sowing), we obtained 443 bacteria strains to analyze bacterial endophytic communities in *V. amurensis* (Figure 1c). These strains were divided into four classes of bacteria: Gammaproteobacteria, Bacilli, Actinobacteria, and Alphaproteobacteria. In general, Gammaproteobacteria was the predominant class, confirming the data from metagenomic analysis (Figure 1a,c). Interestingly, the number of strains of the Bacillus class predominated (67–69%) in grape samples collected near settlements (Ivanovka and Vladivostok) and in the vineyard Makarevich (62%) (Figure 1c). The largest numbers of genera were found in grape samples collected on the Russian Island (Figure 1d). Only one genus, *Bacillus,* was present in all samples from the grapevines, except for the greenhouse grapes (Figure 1d, Appendix A). 

Among the 18 taxa of *V. coignetiae* detected by metagenomic sequencing, sequences of four bacterial classes were the most represented: Alphaproteobacteria (46%), Actinobacteria (25%), Gammaproteobacteria (15%), and Bacteroidia (11%) (Figure 2a). The largest number of taxa from the genus level was represented in wild-growing *V. coignetiae*: 136 and 123 genera for grapes collected in the areas of the cities of Kholmsk and Nevelsk (Figure 2b, Appendix A). More than half of the endophytic bacteria genera (80 genera) were common for all collection points. The 19 unique bacterial genera were endemic for Holmsk, 13 for Nevelsk, and 8 for the Yuzhno–Sakhalinsk Botanical Garden (Figure 2b, Appendix A).

Using a cultivation-dependent approach (bacterial sowing), a total of 102 endophytic bacteria strains were obtained (Figure 2c). These strains were divided into four classes of bacteria: Bacilli, Actinobacteria, Gammaproteobacteria, and Alphaproteobacteria (Figure 2c). The common genera that were detected by bacterial sowing in each *V. coignetiae* sample were *Bacillus* and *Frigoribacterium* (Figure 2d, Appendix A).

The most common taxa for both *V. amurensis* and *V. coignetiae* were *Sphingomonas*, *Methylobacterium*, *Hymenobacter,* and *Comamonadaceae* (Figure 3). The genus *Asinibacterium* was representative only for *V. amurensis* grapevines (Figure 3).

### 2.3. The Biodiversity of Fungal and Fungus-Like Endophytes of Vitis amurensis and Vitis coignetiae Grapevines

Using NGS, we analyzed 841,154 and 220,272 *ITS1* sequences of endophytic fungi and fungus-like organisms from *V. amurensis* and *V. coignetiae*, respectively. The data revealed that the *ITS1* sequences belonged to 180 and 123 taxa from the genus level with the relative representation greater than 0.1% for *V. amurensis* and *V. coignetiae* grapevines, respectively. According to the analysis, these genera belonged to 23 and 22 taxa at the class level in *V. amurensis* and *V. coignetiae*.

The fungal and fungus-like endophytic microbiomes of *V. amurensis* and *V. coignetiae* grapevines differed depending on the grapevine location. The beta diversity analysis showed significant differences between *V. amurensis* and *V. coignetiae* plants performed by PERMANOVA testing (79% of variance, *p* < 0.001) (Appendix A).

Among the 23 classes found in *V. amurensis* plants, sequences of three fungal classes were the most represented: Dothideomycetes—61%, Tremellomycetes—11%, and Malasseziomycetes—8% (Figure 4a). The Malasseziomycetes class was predominant only in *V. amurensis* growing in the greenhouse. The fungal taxonomic diversity at the genus level in grapevines located in the northern regions (Litovko village, Silinsky forest, Verkhneussuriysky Research Station) and on the remote southern Ricord Island was much higher than in other samples (Figure 4b, Appendix A). Twenty-three genera, excluding the plants growing in the greenhouse, were common to all *V. amurensis* samples. It is remarkable that the smallest representation of endophytic fungal and fungus-like genera was detected in the greenhouse *V. amurensis*, and at the same time, four genera were typical only for this sample (Figure 4b, Appendix A).

A cultivation-dependent approach (fungal sowing) was also applied to analyze fun-gal endophytic communities in different grapevines of *V. amurensis*. Unfortunately, no endophytic fungi were found in the four analyzed samples of grapevines (grape from greenhouse, Litovko village, Silinsky forest, SSA, and Ivanovka village). A total of 67 strains were isolated. These strains were divided into six classes of fungi. Dothideomycetes and Tremellomycetes were the predominant classes, confirming the NGS data (Figure 4c). According to the results of fungal sowing, Tremellomycetes and Sordariomycetes fungus classes were present in *V. amurensis* located on Sakhalin Island (Figure 4c). Additionally, Exobasidiomycetes and Leotiomycetes classes were unique for *V. amurensis* grapevines from Ricord Island (Figure 4c). Only one major common genus, *Cladosporium,* was detected for each *V. amurensis* sample, except for plants from Sakhalin and Rikord Islands, according to fungal sowing (Figure 4d, Appendix A).

The most represented fungal classes in *V. coignetiae* grapevines were Dothideomycetes—65% and Tremellomycetes—10% (Figure 5a). Metagenomic analysis also showed the presence of Oomycetes (21%) in the grapevine sample from the botanical garden at Yuzhno–Sakhalinsk, as well as the presence of this class in other samples. Therefore, the Oomycete class was detected as one of the dominant classes in the analysis of *ITS1* sequences in the endophytic fungus-like community (Figure 5a). The 38 taxa from the genus level were common to all samples of *V. coignetiae* (Figure 5b, Appendix A).

Using fungal sowing, 10 strains of endophytic *V. coignetiae* fungi were isolated. Two fungal classes, Dothideomycetes and Tremellomycetes, were present among the analyzed strains by sowing (Figure 5c,d, Appendix A).

The most common fungal taxa for *V. amurensis* and *V. coignetiae* were *Cladosporiom* and *Aureobasidium* (Figure 6). Additionally, the top five genera for *V. amurensis* were *Malassezia*, *Vishniacozyma,* and *Kabatina*. The genera *Ramularia*, *Plasmopara,* and *Vishniacozyma* were predominant for *V. coignetiae*. The genus *Saccharomyces* was detected only in *V. amurensis* (Figure 6).

### 2.4. Comparative Analysis of Endophytic Microbial Communities in Vitis amurensis, Vitis coignetiae, and Vitis vinifera

We performed a comparative analysis of the endophytic communities of *V. amurensis* and *V. coignetiae* obtained in this study with the previously studied endophytic communities of *V. vinifera* growing in Germany [39] and the USA (California) [21]. The main criteria for the selection of these metagenomic studies were the presence of a microorganism film removal step in the process of endophyte isolation, the same bacterial *16S* rRNA and fungal *ITS1* rDNA region (*ITS1*–*ITS2*), and Illumina sequencing technology. The samples used in the comparative analysis are presented in Appendix A.

Total of 10,415,438 and 1,152,380 *16S* paired-end reads were obtained from California’s and Germany’s *V. vinifera* samples in NCBI BioProject database, respectively. After paired-end alignments and filtering procedures, totals of 470,604 and 5866 *16S* sequences were generated from California’s 44 samples and Germany’s 15 samples, respectively (Appendix A).

The amplicon data of each sample site with respect to the location were analyzed. The results for alpha and beta bacterial endophytic diversity analysis are shown in Figure 7c,d, respectively. The bacterial biodiversity of grape samples *V. amurensis*, *V. coignetiae,* and *V. vinifera* were similar in alpha diversity based on Shannon diversity index (Figure 7c). The beta diversity results are presented in the nonmetric multidimensional scaling (NMDS) ordination (Figure 7d). The PERMANOVA test demonstrated that there is a significant proportion of variance among the *V. amurensis*, *V. coignetiae*, *V. vinifera* cv. ‘Syrah’, and *V. vinifera* cv. ‘Cabernet Dorsa’ microbial communities (29% of variance, *p* < 0.001). NMDS ordination showed that the samples of *V. amurensis* and *V. coignetiae* were more similar based on beta diversity than the samples from California and Germany. (Figure 7d).

According to the analysis, 31 bacterial taxa from the class level were detected in *V. amurensis*, *V. coignetiae,* and *V. vinifera*. The most represented endophytic bacteria for all analyzed samples of grapevines belonged to the Gammaproteobacteria, Alphaproteobacteria, Actinobacteria, Bacteroidia, and Bacilli classes (Figure 7a). For *V. amurensis* and *V. vinifera*, the dominant class of endophytic bacteria was identified as Gammaproteobacteria (35–47%), while for *V. coignetiae* samples, the dominant class was Alphaproteobacteria (46%) (Figure 7a). The leaders in the biodiversity of endophytic bacteria were *V. amurensis* and *V. vinifera* from California (Figure 7b). Notably, a total of 74 common genera of endophytic bacteria were detected in all analyzed grapevines (Figure 7b, Appendix A). There were 25 genera of bacterial endophytes characteristic only for the Far Eastern grapes of Russia *V. amurensis* and *V. coignetiae*. There were no unique bacterial genera in *V. coignetiae* samples, while four unique bacterial genera were found for *V. amurensis*—*Phascolarctobacterium*, *Candidatus Cardinium*, [*Ruminococcus*] torques group and *Ampullimonas* (Figure 7b, Appendix A).

For the comparative analysis of endophytic *ITS1* communities in grapevines, a total of 433,828 paired-end reads were obtained from California’s *V. vinifera* cv. ‘Syrah’ samples in the NCBI BioProject database. After paired-end alignments and filtering procedures, a total of 267,626 *ITS1* sequences were generated from 29 California’s samples (Appendix A).

The results for alpha and beta diversity analysis of fungi and fungus-like endophytic biodiversity in the different grape samples are shown in Figure 8c,d, respectively. The *V. amurensis* and *V. coignetiae* grape samples from Russia were statistically different from *V. vinifera* cv. ‘Syrah’ in alpha diversity. Shannon’s diversity indices for *V. amurensis* and *V. coignetiae* samples were higher than that for *V. vinifera* cv. ‘Syrah’ samples. Additionally, the Shannon diversity index was slightly higher in *V. amurensis* than in *V. coignetiae* samples (Figure 8c). NMDS ordination showed that samples of *V. amurensis*, *V. coignetiae*, and *V. vinifera* cv. ‘Syrah’ were located in separate clusters, while *V. amurensis* and *V. coignetiae* samples were more similar than *V. amurensis* and *V. vinifera* cv. ‘Syrah’ samples (Figure 8d). The PERMANOVA test demonstrated that the samples of *V. amurensis*, *V. coignetiae,* and *V. vinifera* cv. ‘Syrah’ were significantly different based on beta diversity (38% of variance, *p* < 0.001) (Figure 8d). According to the NMDS ordination plot, the samples of *V. amurensis* and *V. coignetiae* were similar to but significantly different from California’s samples.

According to the comparative analysis, the grape endophytic mycobiome was represented by 24 main classes of fungi and fungus-like organisms (Figure 8a). The most prevalent classes for all grapevines were Dothideomycetes (61–87%) and Tremellomycetes (4–11%). The third most predominant class of endophytic fungi or fungus-like microorganisms for wild *V. amurensis* was Malasseziomycetes (8%); for wild *V. coignetiae*, it was Oomycetes (9%); and for *V. vinifera* cv. ‘Syrah’, it was Cystobasidiomycetes (5%) (Figure 8a). The largest numbers of genera of micromycetes were found in *V. amurensis* and *V. coignetiae:* 180 and 123, respectively (Figure 8b, Appendix A). All samples of grapes had 30 intersecting genera, while *V. amurensis* and *V. coignetiae* had 89 intersecting genera, and *V. amurensis* and *V. vinifera* cv. ‘Syrah’ had only 6 (Figure 8b, Appendix A). The mycobiome of *V. amurensis* was represented by 55 unique fungal genera, while the mycobiome of *V. coignetiae* was represented by four unique genera (Figure 8b, Appendix A).

## 3. Discussion

Microbiomes have been found to be critical for plants, promoting growth, elevating stress tolerance, and expanding the plant’s metabolic repertoire with novel defense pathways [40]. Microbial resources, with the advantages of economic efficiency, harmlessness to the environment, and non-toxicity to organisms, have been recommended as promising alternatives to conventional fertilizers and pesticides [41]. Additionally, the complete characterization of endophytes may help plant adaptation to drought, heat, and heavy metal stress, as demonstrated for wheat [42], brassica [43], and other plant species [44]. There has also been longstanding interest in characterizing the biodiversity of microbial endophytes from wild plants. Wild plants and weeds largely have a richer variety of endophytic microorganisms contributing to plant survival under adverse conditions. It is believed that domesticated plants may be colonized by distinct endophytic microbes, which might provide plant hosts with special traits for adaptability and survival under adverse environmental conditions [45]. For example, it was shown that bacterial endophytes isolated from the roots and soil of healthy wild plants of *Plantago ovata*, *Albizia lebbeck*, *Astragalus spinosus,* and others from Saudi Arabia could promote plant growth by developing siderophores, solubilizing phosphate, synthesizing indole 3 acetic acid (IAA), and inhibiting the growth of some plant fungal pathogens, such as *Fusarium*, *Rhizoctonia solani*, *Botrytis*, and *Stemphylium* spp. [46]. Additionally, it was shown that the endophytic bacteria *Microbacteriaceae bacterium*, *Microbacterium testaceum,* and *Bacillus subtilis* from wild rice exhibit plant growth promoting activity with high efficiency and can form a prospective consortium, which can be employed as a source of bio-fertilizers for the enhancement of plant growth and development [47]. A *Diaporthe* fungal endophyte from a wild grass, *Festuca rubra,* improves the growth and salinity tolerance of domesticated grasses [48]. In addition, endophytes can be considered a source of bioactive compounds, such as alkaloids, benzopyranones, chinones, flavonoids, phenolic acids, quinones, steroids, terpenoids, tetralones, and xanthones [49]. Thus, endophytic microbes from wild plants have great potential for application in agriculture and represent an important source of various compounds with valuable biological activities.

This study is the first comprehensive effort to characterize a group of endophytic bacteria, fungi, and fungus-like microorganisms associated with asymptomatic tissue from the wild grapevines *V. amurensis* and *V. coignetiae* of the Russian Far East. The results of the present study pointed out that the endophytic microbes of wild *V. amurensis* are very diverse, and the microbial diversity depends on the grapevine location. The biodiversity of endophytic bacteria from *V. amurensis* was represented by 114–177 genera, whereas only one-third was common for all samples of the Amur grape (53 genera). We found that *V. coignetiae* was characterized by a smaller variety of endophytic bacterial genera (105–136 genera), whereas more than a half (80 genera) was common for all analyzed samples of *V. coignetiae*. Similar results have been obtained when analyzing the biodiversity of endophytic micromycetes and fungus-like microorganisms from *V. amurensis* and *V. coignetiae*. The mycobiome of *V. amurensis* was represented by 59–114 genera, whereas only 23 genera were found to be common for all samples of *V. amurensis*. For *V. coignetiae*, the number of fungi and fungus-like genera in each sample varied from 75 to 81, whereas half of them (38 genera) were common to all samples. Perhaps the large number of common endophytic genera of bacteria and fungi of *V. coignetiae* was due to the island localization of this grape species. We also noted that the biodiversity of endophytic bacteria of *V. amurensis* strongly depended on the location of sample collection. Considering samples collected on the islands or in the northern regions, the dominant class of endophytic bacteria changed to the Alphaproteobacteria class, which was dominant in *V. coignetiae* samples from Sakhalin Island. It is possible that anthropogenic factors considerably changed the grapevine endophytic community. The samples collected on the island territories exhibited a richer endophytic diversity of both bacteria and fungi. Additionally, we found that the wild grapevines *V. amurensis* and *V. coignetiae* exhibited a unique composition of the endophytic bacterial and fungal communities, which was significantly different from the microbial composition of the cultivated *V. vinifera*. It is possible that the endophytic biodiversity of grapes in areas remote from human activity was higher due to the lack of chemical plant treatments, and this diversity was associated with the natural development of the endophytic microflora of grapes, which has been formed for decades. It was previously shown that endophytic communities were strongly affected by foliar age and domestication, which were crucial factors in establishing symbiotic associations with selective enrichment for specific endophytes [50]. Differences in the endophyte preferences of *V. amurensis* or *V. coignetiae* were probably related to the physiological characteristics of grape species, such as pH, sugar content, secondary metabolites, etc.

According to the biodiversity of endophytic fungi and fungus-like microorganisms in *V. amurensis*, the samples taken from the greenhouse and the botanical garden on Sakhalin Island were very different. It is likely that the strong difference in the biodiversity of endophytic fungi was associated with the soil on which the grapevine was grown. Commercial soil was used to cultivate *V. amurensis* in the greenhouse, and chemical plant treatments may have been carried out at the Botanical Garden on Sakhalin Island. Previously, it was shown that addition of chemical fertilizers and pesticides reduced the amount and diversity of arbuscular mycorhizae in plants [51,52]. We also noted a large percentage of class Oomycetes in *V. coignetiae* samples. According to the comparative analysis of the generic composition of the *V. amurensis* and *V. coignetiae* mycobiomes, the genus *Plasmopara* was represented by 1.5% and 8%, respectively. Perhaps the presence of *Plasmapara* was due to the wild natural area without specialized treatment, the island location of *V. coignetiae* samples, and the specifics of the climate on Sakhalin Island. The discovered Oomycetes might be non-pathogenic for the wild plants but cause diseases for the human-cultivated varieties. The plant-associated microbiota contributed to maximizing host adaptation to pathogenic infection. For example, the grape endophytes, including *Bacillus*, *Variovorax*, *Pantoea*, *Staphylococcus*, *Herbaspirillum*, or *Sphingomonas* strains, were shown to inhibit mycelial growth of *Phytophthora infestans*, which the authors used as a surrogate for *P. viticola* [53]. Therefore, a detailed study of the endophytic diversity of grape samples with different representation of Oomycetes could reveal microorganisms that are associated with oomycetes or are antagonists of oomycetes, *Plasmapara* in particular, but this assumption requires further research.

The genus of endophytic bacteria *Asinibacterium* was unique to the wild *V. amurensis* grapes. It is known that *Asinibacterium* bacteria are able to grow under elevated levels of heavy metals [54]. It is possible that *Asinibacterium* strains isolated from the wild grapevines can be used in agriculture. Additionally, it was found that the genus *Saccharomyces* was representative only for the *V. amurensis* endophytic fungal community. It is known that yeasts of the genus *Saccharomyces* regulate the acidity of wines due to malic acid production [55]. In addition, application of endophytic yeast leads to an increase in lactic acid content compared to selected yeast strains [56]. It is known that the malolactic fermentation is of the utmost importance for wine quality. Moreover, application of spontaneous grape microbiota leads to incomplete fermentation of sugars, thus leading to sweeter wines compared to the fermentation with selected strains [56]. This difference is important, especially in the case of artisanal wines (or those produced in the household). It is likely that the endophytic yeast of wild *V. amurensis* will be of interest to winemakers to preserve wine acidity or to produce sweeter wines when applying selected microorganisms during alcoholic fermentation, which can be a promising solution to preserve wine quality and to create wines with new taste qualities.

In addition, the analysis of the endophytic community biodiversity for *V. amurensis* and *V. coignetiae* grapes detected microorganisms that were present in the human microbiome, such as the bacterial genera *Lactobacillus*, *Enterococcus*, *Bacteroides*, *Streptococcus*, *Escherichia*-*Shigella*, and *Bacillus* and the yeast *Saccharomyces* (Appendix A). The potential of probiotic species has recently motivated researchers to examine the production of probiotic foods and the modulation of the gut microbiota. Previously, it was shown that several *Lactobacillus* spp. and other coliform bacteria were the most promising probiotic species with a considerable role in the prevention of degenerative diseases, such as obesity, diabetes, cancer, cardiovascular diseases, malignancies, liver disease, and inflammatory bowel disease [57]. Additionally, a recent study showed that *Saccharomyces* spp. prevented inflammation by the promotion of proinflammatory immune function via the production of short-chain fatty acids [57]. Thus, eating grapes with a certain endophytic composition is a valuable source of probiotics that can have beneficial effects on human health.

In conclusion, the obtained data on the endophytic communities in wild grapes *V. amurensis* and *V. coignetiae* are important for the selection of perspective strains of endophytic microorganisms for use as potential biological agents in plant disease protection and plant functional studies. The results of the present study are important for agriculture, winemaking, and plant biotechnology in terms of obtaining new strains that produce biologically active compounds.

## 4. Materials and Methods

### 4.1. Plant Material and Sampling

This study used asymptomatic tissues (young shoots with three leaves) of 11 10- to 15-year-old grapevines of *V. amurensis* and three grapevines of *V. coignetiae* collected in July 2022. *V. amurensis* (Gh) was cultivated in greenhouse conditions at the Laboratory of Biotechnology, Federal Scientific Center of the East Asia Terrestrial Biodiversity, Far Eastern Branch of the Russian Academy of Sciences, Vladivostok, Russia. *V. amurensis* (M) was located at the commercial vineyard “Makarevich”. *V. amurensis* (S-Va) was sampled at the botanical garden on Sakhalin Island. The other eight *V. amurensis* grapevines were sampled from several nonprotected natural populations: two grapevines located at a distance of 1 km from each other near Vladivostok, Russia (P1 and P2); two grapevines located on Russky and Rikord Islands, the southern Primorsky Territory of the Russian Far East (P3 and P4); two grapevines in Ivanovka village (P5) and the Verkhne–Ussuriysky Research Station (SSA) of the Federal Scientific Center of the East Asia terrestrial biodiversity Far East Branch of the Russian Academy of Sciences (P6); and two grapevines located in Litovko village (Kh-1) and Silinsky forest (Kh-2) of the southern Khabarovsk region of the Russian Far East. One *V. coignetiae* grapevine was sampled from the botanical garden of Sakhalin Island (S-1). Two other *V. coignetiae* grapevines were located in a nonprotected natural population on Sakhalin Island: near the city of Kholmsk (S-2) and the city of Nevelsk (S-3) (Table 1).

The values of the average temperatures and precipitation in the southern Primorsky Territory of the Russian Far East (M, P-1,P-2, P-3, P-4) in July 2022 were 20 °C and 150 mm, respectively (http://www.pogodaiklimat.ru/history/31960.htm, accessed on 1 March 2023). In the central part of Primorsky Territory (P-5, P-6), the average temperature and precipitation in July 2022 were 23 °C and 130 mm (http://www.pogodaiklimat.ru/history/31981_2.htm, accessed on 1 March 2023). The values of the average temperatures and precipitation in the southern Khabarovsk region of the Russian Far East (Kh-1, Kh-2) in July 2022 were 23 °C and 75 mm (http://www.pogodaiklimat.ru/history/31960.htm, accessed on 1 March 2023). On Sakhalin Island (S-Va, S-1, S-2, S-3), the average temperature and precipitation were 18 °C and 50 mm (http://www.pogodaiklimat.ru/history/32128_2.htm, accessed on 1 March 2023). Each plant material specimen was delivered to the laboratory in sterile bags within 1–2 days.

The plant material was collected in July for both the genomic approaches and the classical microbiology techniques (bacterial and fungal sowing). Two biological replicates (stems and leaves) of each grapevine were collected. In total, 22 biological replicates of *V. amurensis* and six biological replicates of *V. coignetiae* were collected and analyzed by the cultivation-dependent approach. For the cultivation-independent approach (NGS), we used the grapevine material collected in July (a total of 22 biological replicates of *V. amurensis* and six biological replicates of *V. coignetiae*) and applied two technical replications per biological replicate.

### 4.2. Isolation and Identification of the Endophytic Bacteria and Fungi

The grapevine tissues (1.5 g) were surface-sterilized according to the method described earlier [58,59]. To check the efficacy of this method of surface sterilization, 100 µL of the last wash water were incubated on R2A (PanReac, AppliChem, Darmstadt, Germany) and on potato dextrose agar (PDA, Neogene, Watford (Hertfordshire), UK) plates to check for the absence of colony growth from the outside. The surface-sterilized tissues of *V. amurensis* and *V. coignetiae* were ground to a homogeneous mass in a sterile mortar; the resulting juice was squeezed, and a 100-μL aliquot was transferred to R2A (for bacteria) and PDA (for fungi) plates. After 2 and 7 days, the grown bacteria and fungal colonies, respectively, were sampled and carefully transferred to a new sterile Petri dish for repeated cultivation. A total of 545 separate strains of endophytic bacteria and 77 separate strains of endophytic fungi were obtained.

DNA from the separate strains of endophytes was isolated by the hexadecyltrimethylammonium bromide (CTAB) method with modifications [60] Bacterial *16S* rRNA gene sequences were amplified using universal bacterial primers for the amplification of approximately 1500-bp *16S* PCR products (8F, 5′AGA GTT TGA TCM TGG CTC AG and 1522R, 5′AAG GAG GTG ATC CAR CCG CA) [61]. Fungal *ITS1* rDNA were amplified using universal primers for the amplification of approximately 560-bp *ITS1* PCR products (5′AGG AGA AGT CGT AAC AAG G and 5′TCC TCC GCT TAT TGA TAT GC) [62]. PCR products were sequenced using an ABI 3130 Genetic Analyzer (Applied Biosystems, Foster City, CA, USA) according to the manufacturer’s instructions as described [63]. The Basic Local Alignment Search Tool (BLAST) program was used for sequence analysis. Multiple sequence alignments were performed using the Clustal X program [64]. A sequence identity of ≥99% was considered a sufficient threshold value for taxonomic identification of bacterial genera.

### 4.3. DNA Extraction, Library Preparation, and Illumina MiSeq Sequencing

For NGS, the DNA was isolated from leaves and stems of *V. amurensis* and *V. coignetiae* using the CTAB-spin method described earlier [58,59,65]. The DNA samples were sent to Sintol (Moscow, Russia) for high-throughput sequencing using Illumina technology. DNA was assessed for quality and quantity using the Nanodrop-1000 (Thermo Fisher Scientific, Waltham, MA, USA) and Quantus Fluorometer (Promega, Madison, WI, USA), respectively. The libraries were prepared for sequencing according to the protocol described in the manual “*16S* Metagenomic Sequencing Library Preparation” (Part # 15,044,223 Rev. B; Illumina). Bacterial *16S* rRNA regions were amplified from all samples using the modified for *Vitis* sp. plants primers 515F (5′GGTAATACGKAGGKKGCDAGC) and 806R (5′RTGGACTACCAGGGTATCTAA). Fungal *ITS1* rDNA regions were amplified from all samples using the primers ITS1f (5′CTTGGTCATTTAGAGGAAGTAA) and ITS2 (5′ GCTGCGTTCTTCATCGATGC). Amplicons were indexed using the Nextera^®^ XT Index Kit reagent kit (Illumina, San Diego, CA, USA). The library pool was sequenced on Illumina MiSeq (2 × 250 paired end) using MiSeq Reagent Kit v2 (500 cycles).

Bacterial and fungal endophyte sequences were deposited in NCBI under the accession number PRJNA980748 and in the database of the laboratory of biotechnology, Federal Scientific Center of the East Asia Terrestrial Biodiversity, Far Eastern Branch of the Russian Academy of Sciences, Russia (https://biosoil.ru/downloads/biotech/Vitis%20metagenom/2022-08-seq=Illumina=3 (accessed on 19 July 2023) and https://biosoil.ru/downloads/biotech/Vitis%20metagenom/2022-11-seq=Illumina=4/ (accessed on 19 July 2023)).

### 4.4. Data Processing

The samples used in the bioinformatic analysis are presented in Appendix A. The obtained data were processed using custom scripts in the R and Bash languages (https://github.com/niknit96/Aleynova_et.al.2023/, accessed on 19 July 2023). Raw readings were preprocessed using the QIIME 2 [66] and DADA2 [67] programs. The primers, remaining PhiX reads, and chimeric sequences were removed, and paired-end reads were merged and sorted. Taxonomic identification of sequences was performed using the QIIME 2 Scikit-learn algorithm using the SILVA 138 pre-trained classifier for *16S* sequences (99% OTUs from V4 region of sequences) [68] and the UNITE pre-trained classifier for *ITS* sequences (99% OTUs from ITS1f/ITS2 region of sequences) [69].

The qiime2R [70], phyloseq [71], ggdendro [72], RColorBrewer [73], circlize [74], and tidyverse [75] libraries were used in pre-filtering and data preparation. Amplicon sequence variants were merged into genus-level taxonomic ranks. Mitochondria, chloroplast, *Viridiplantae,* and unidentified sequences were deleted from the obtained data. Genus-level taxa were filtered based on a relative abundance of >0.1% for a plant. In the class-level taxonomy bar plots, we merged the filtered genera to one group called “other”. Additionally, “other” genus taxa were removed from the UpSet diagrams. Shannon alpha diversity and Bray–Curtis beta diversity data were obtained using the Vegan package [76]. Bray–Curtis dissimilarity data were transformed into even sampling depths and converted to nonmetric multidimensional scaling (NMDS). Wilcoxon’s pairwise rank sum test with the false discovery rate correction method was performed to analyze the alpha diversity data between groups. Statistical validation of beta diversity data was performed using the PERMANOVA test with 999 permutations. The ggplot2 [75] and ComplexHeatmap [77] R libraries were used in the graphical representation of the results.

## Figures and Tables

**Figure 1 plants-12-02952-f001:**
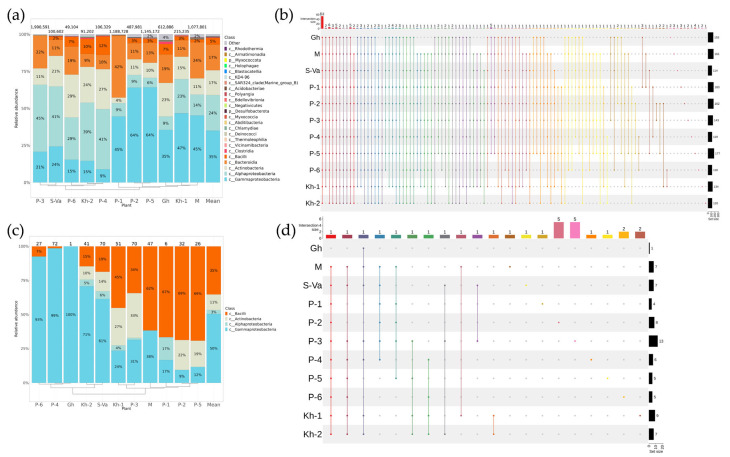
Composition of endophytic bacterial communities in wild grape *Vitis amurensis* according to genomic approach (next-generation sequencing (NGS)) and classical microbiology technique (bacterial sowing (Sow)). The composition of endophytic bacteria of *V. coignetiae* depends on the grapevine location: Gh—greenhouse at the Laboratory of Biotechnology, Federal Scientific Center of the East Asia Terrestrial Biodiversity; M—the commercial vineyard “Makarevich”; S-Va—the botanical garden on Sakhalin Island; P-1—Vladivostok; P-2—Vladivostok; P-3—Russky Island; P-4—Rikord Island; P-5—Ivanovka village; P-6—the Verkhne–Ussuriysky Research Station (SSA); Kh-1—the Litovka village, the southern Khabarovsk region of the Russian Far East; Kh-2—the Silinsky forest; Mean—mean value between plants. (**a**) Class-level taxonomical bar plots for the bacterial endophytic communities of *V. amurensis* from different locations and the mean data obtained using NGS; (**b**) genus-level UpSet diagrams depicting overlapping taxa of NGS for grapevine samples from different locations; (**c**) class-level taxonomical bar plots for the bacteria endophytic communities of *V. amurensis* from different locations and the mean data obtained using bacterial Sow; (**d**) genus-level UpSet diagrams depicting overlapping taxa of bacterial Sow for grapevine samples from different locations. Genus-level taxa were filtered based on relative abundance of >0.1% for a plant. Genera of relative abundance of <0.1% were removed from UpSet diagram. Number of sequences or strains (for Sow) are shown above in taxonomical bar plots. For clustering in bar plots, we used the unweighted pair group method with arithmetic mean (UPGMA).

**Figure 2 plants-12-02952-f002:**
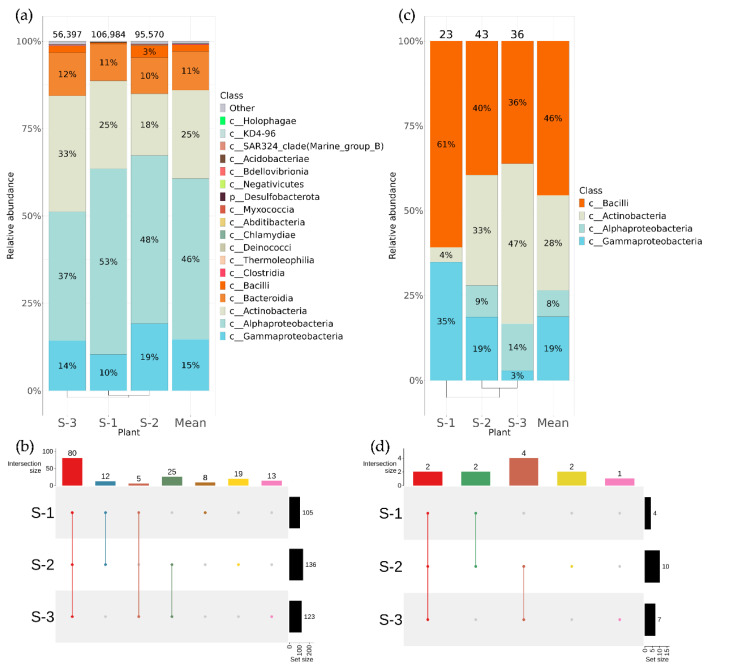
Composition of endophytic bacterial communities in wild grape *Vitis coignetiae* according to genomic approach (next-generation sequencing (NGS)) and classical microbiology technique (bacterial sowing (Sow)). The composition of endophytic bacteria of *V. coignetiae* depends on the grapevine location: S-1—the botanical garden on Sakhalin Island; S-2—near the city of Kholmsk on Sakhalin Island; S-3—near the city of Nevelsk on Sakhalin Island; Mean—mean value between plants. (**a**) Class level taxonomical bar plots for the bacteria endophytic communities of *V. coignetiae* from different locations and the mean data obtained using NGS; (**b**) genus-level UpSet diagrams depicting overlapping taxa of NGS for *V. coignetiae* grapevine samples from different locations; (**c**) class-level taxonomical bar plots for the bacteria endophytic communities of *V. coignetiae* from different locations and the mean data obtained using bacterial sow; (**d**) genus-level UpSet diagrams depicting overlapping taxa of bacterial sow for *V. coignetiae* grapevine samples from different locations. Genus level taxa were filtered based on relative abundance of >0.1% for a plant. Genera of relative abundance of <0.1% were removed from the UpSet diagram. Number of sequences or strains (for sow) are shown above in taxonomical bar plots. For clustering in bar plots, we used UPGMA.

**Figure 3 plants-12-02952-f003:**
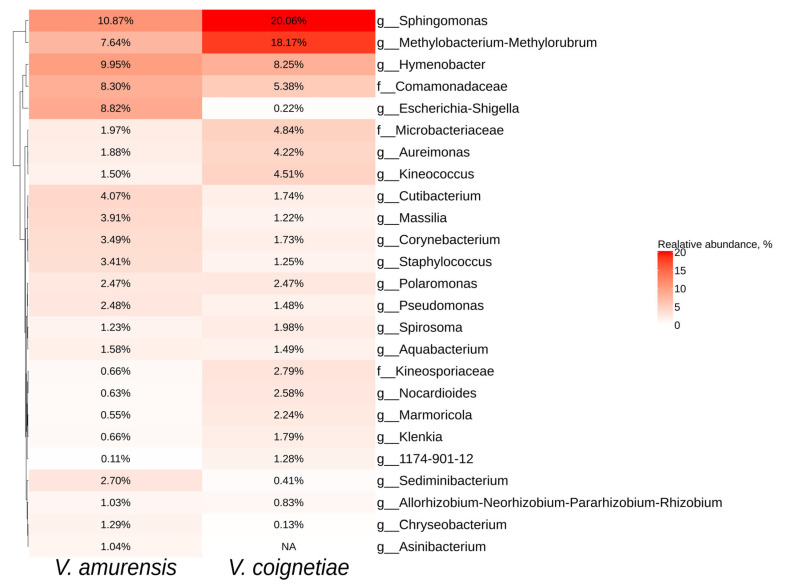
Heatmap of the relative abundances of endophytic bacteria of significant taxa at the genus level according to next-generation sequencing (NGS) in *Vitis amurensis* and *Vitis coignetiae*. The top 20 most abundant taxa from each factor are displayed. White squares (NA) represent the absence of taxa.

**Figure 4 plants-12-02952-f004:**
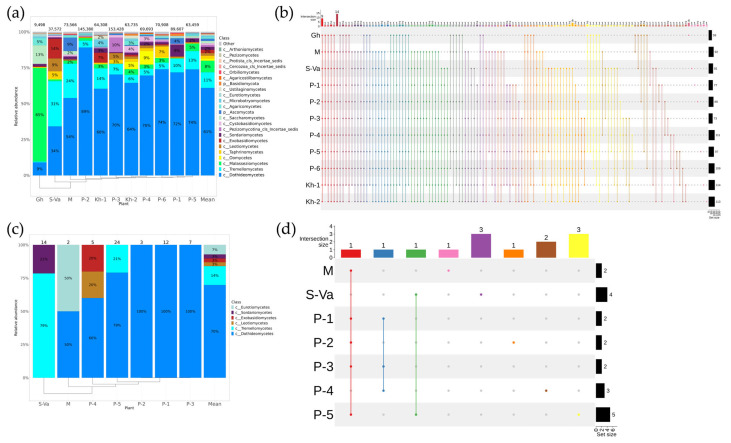
Composition of the fungal and fungus-like endophytic community in wild grape *Vitis amurensis* according to genomic approach (next-generation sequencing (NGS)) and classical microbiology technique (bacterial sowing (Sow)). The composition of endophytic bacteria of *V. coignetiae* depends on the grapevine location: Gh—greenhouse at the Laboratory of Biotechnology, Federal Scientific Center of the East Asia Terrestrial Biodiversity; M—the commercial vineyard “Makarevich”; S-Va—the botanical garden on Sakhalin Island; P-1—Vladivostok; P-2—Vladivostok; P-3—Russky Island; P-4—Rikord Island; P-5—Ivanovka village; P-6—the Verkhne-Ussuriysky Research Station (SSA); Kh-1—Litovko village, the southern Khabarovsky region of the Russian Far East; Kh-2—the Silinsky forest; Mean—mean value between plants. (**a**) Class-level taxonomical bar plots for the fungal endophytic communities of *V. amurensis* from different locations and the mean data obtained using NGS; (**b**) genus-level UpSet diagrams depicting overlapping taxa of NGS for *V. amurensis* grapevine samples from different locations; (**c**) class-level taxonomical bar plots for the fungal endophytic communities of *V. amurensis* from different locations and the mean data obtained using fungal sow; (**d**) genus-level UpSet diagrams depicting overlapping taxa of fungal sow for *V. amurensis* grapevine samples from different locations. Genus-level taxa were filtered based on relative abundance of >0.1% for a plant. Genera of relative abundance of <0.1% were removed from the UpSet diagram. Number of sequences or strains (for sow) are shown in above taxonomical bar plots. For clustering in bar plots, we used UPGMA.

**Figure 5 plants-12-02952-f005:**
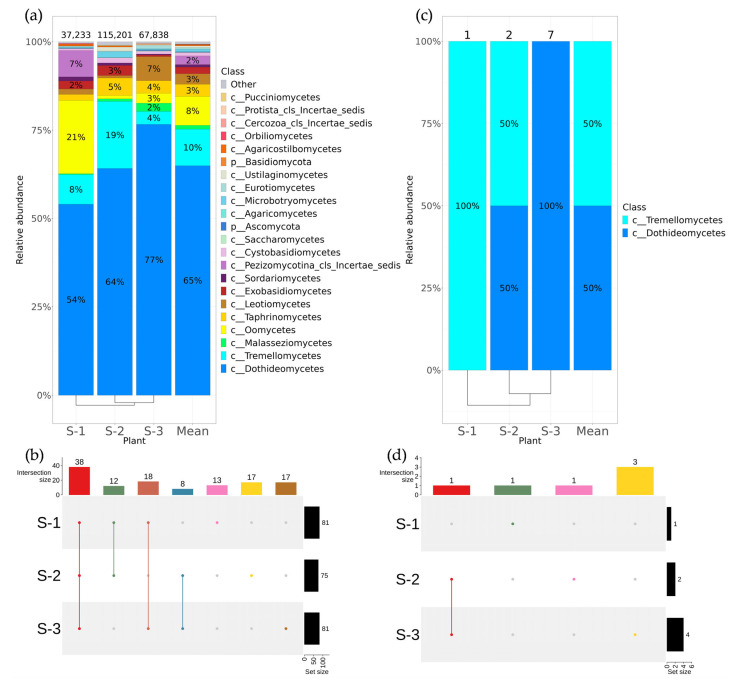
Composition of the fungal and fungus-like endophytic communities in wild grape *Vitis coignetiae* according to genomic approach (next-generation sequencing (NGS)) and classical microbiology technique (fungal sowing (Sow)). The composition of endophytic fungi of *V. coignetiae* depends on grapevine locations: S-1—the botanical garden on Sakhalin Island; S-2—near the city of Kholmsk on Sakhalin Island; S-3—near city Nevelsk of Sakhalin Island; Mean—mean value between plants. (**a**) Class-level taxonomical bar plots for the fungal endophytic communities of *V. coignetiae* from different locations and the mean data obtained using NGS; (**b**) genus-level UpSet diagrams depicting overlapping taxa of NGS for *V. coignetiae* grapevine samples from different locations; (**c**) class-level taxonomical bar plots for the fungal endophytic communities of *V. coignetiae* from different locations and the mean data obtained using fungal Sow; (**d**) genus-level UpSet diagrams depicting overlapping taxa of fungal Sow for *V. coignetiae* grapevine samples from different locations. Genus-level taxa were filtered based on relative abundance of >0.1% for plant. Genera of relative abundance of <0.1% were removed from the UpSet diagram. Number of sequences or strains (for Sow) are shown in above taxonomical bar plots. For clustering in bar plots, we used UPGMA.

**Figure 6 plants-12-02952-f006:**
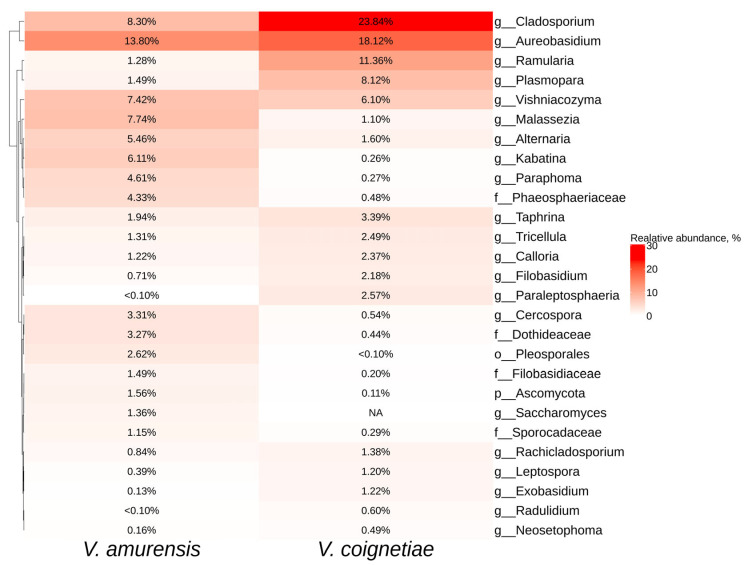
Heatmap of the relative abundance of endophytic fungi of significant taxa at the genus level according to next-generation sequencing (NGS) in *Vitis amurensis* and *Vitis coignetiae*. The top 20 most abundant taxa from each factor are displayed. White squares (NA) represent the absence of taxa.

**Figure 7 plants-12-02952-f007:**
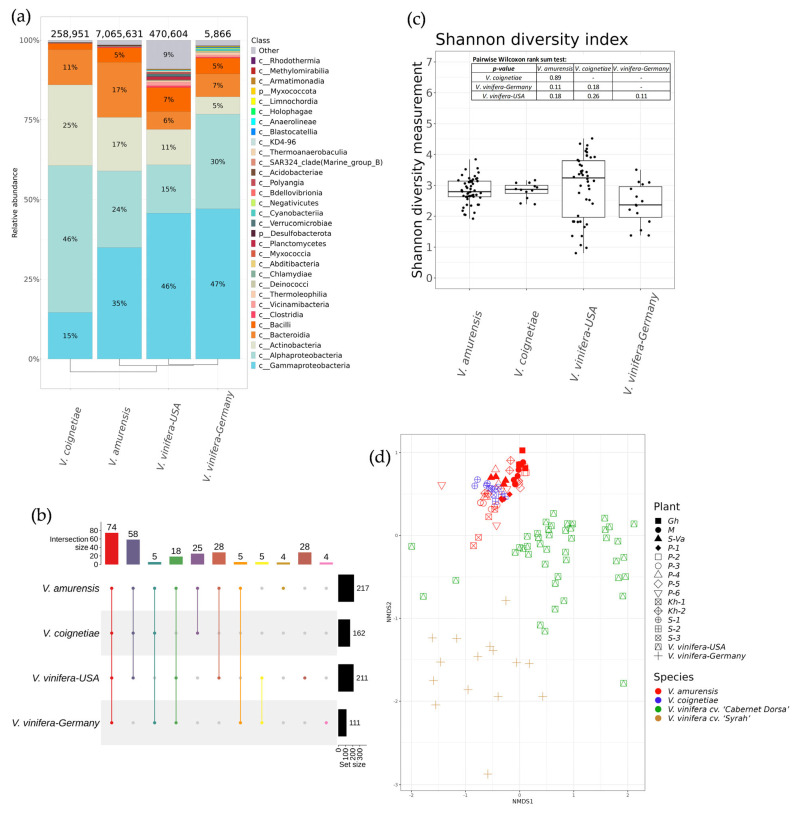
A comparison of endophytic bacterial communities of wild *Vitis amurensis* and *Vitis coignetiae* from the Russian Far East with the endophytic community of cultivated *Vitis vinifera* cv. ‘Syrah’ from the USA and *V. vinifera* cv. ‘Cabernet Dorsa’ from Germany. (**a**) Class-level taxonomical bar plots for the endophytic bacterial community of *V. amurensis*, *V. coignetiae*, *V. vinifera* cv. ‘Syrah’ from USA, and *V. vinifera* cv. ‘Cabernet Dorsa’ from Germany; (**b**) genus-level UpSet diagrams depicting overlapping taxa in samples; (**c**) Shannon′s alpha diversity boxplot; (**d**) Bray–Curtis beta diversity NMDS plot. Genus-level taxa were filtered based on relative abundance of >0.1% for a plant. Genera of relative abundance of <0.1% were removed from the UpSet diagram. For clustering in bar plots, we used UPGMA.

**Figure 8 plants-12-02952-f008:**
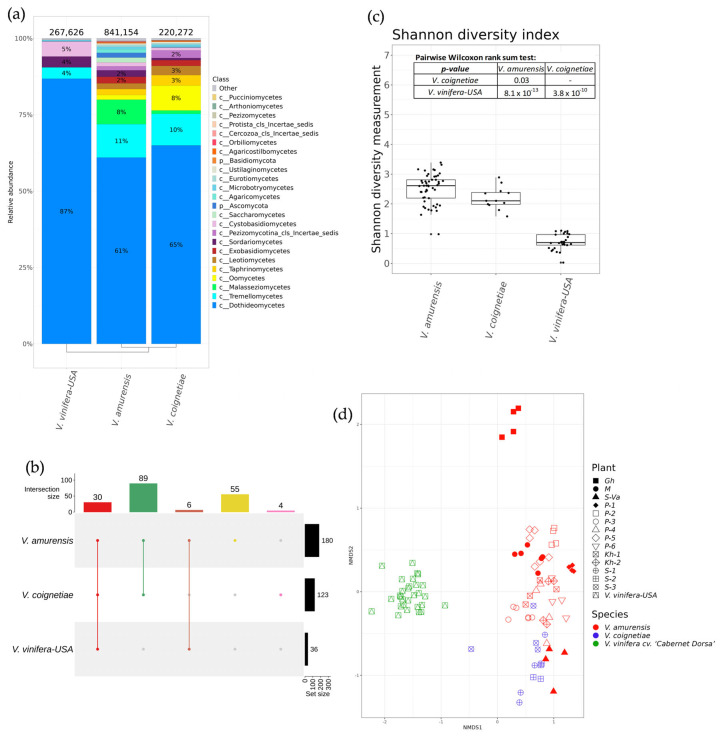
A comparison of fungi and fungus-like endophytic communities of wild *Vitis amurensis* and *Vitis coignetiae* from the Russian Far East with the endophytic community of cultivated *Vitis vinifera* cv. ‘Syrah’ from the USA (California) (**a**) Class-level taxonomical bar plots for the fungi and fungus-like endophytic community of *V. amurensis*, *V. coignetiae*, and *V. vinifera* cv. ‘Syrah’ from the USA (California)*;* (**b**) Shannon′s alpha diversity boxplot; (**c**) genus-level UpSet diagrams depicting overlapping taxa in samples; (**d**) Bray–Curtis beta diversity NMDS plot. Genus-level taxa were filtered based on relative abundance of >0.1% for a plant. Genera of relative abundance of <0.1% were removed from the UpSet diagram. For clustering in bar plots, we used UPGMA.

**Table 1 plants-12-02952-t001:** The abbreviations, locations, and map coordinates of samples of grapevines of *Vitis amurensis* and *Vitis coignetiae* collected in July 2022. Abb.—abbreviation; Map—map coordinates.

No	Abb.	Location	Map
*Vitis amurensis*			
1	Gh	Greenhouse at the Laboratory of Biotechnology, Federal Scientific Center of the East Asia Terrestrial Biodiversity, Far Eastern Branch of the Russian Academy of Sciences, Vladivostok, Russia	43.192929, 131.921064
2	M	The commercial vineyard “Makarevich”	43.718600, 132.11040
3	S-Va	The botanical garden on Sakhalin Island	46.943386, 142.758662
4	P-1	Vladivostok, the southern Primorsky Territory of the Russian Far East	43.2242327, 131.991123
5	P-2	Vladivostok, the southern Primorsky Territory of the Russian Far East	43.2242327, 131.991123
6	P-3	Russky Island, the southern Primorsky Territory of the Russian Far East	42.964116, 131.881337
7	P-4	Rikord Island, the southern Primorsky Territory of the Russian Far East	42.870564,131.655779
8	P-5	Ivanovka village, the center of Primorsky Territory of the Russian Far East	43.975818, 132.481537
9	P-6	The Verkhne–Ussuriysky Research Station (SSA) of the Federal Scientific Center of the East Asia Terrestrial Biodiversity, Far Eastern Branch of the Russian Academy of Sciences, the center of Primorsky Territory of the Russian Far East	44.034842, 134.202996
10	Kh-1	Litovko village, the southern Khabarovsk region of the Russian Far East	49.2540051, 135.1847371
11	Kh-2	Silinsky forest, the southern Khabarovsk region of the Russian Far East	50.586259, 137.0172029
*Vitis coignetiae*			
1	S-1	The botanical garden on Sakhalin Island	46.943386, 142.758662
2	S-2	Near the city of Kholmsk on Sakhalin Island	46.982750, 142.117957
3	S-3	Near the city of Nevelsk on Sakhalin Island	46.658645, 141.888784

## Data Availability

The data presented in this study are available within the article and Appendix A.

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
