# Peer review of "The Endophytic Microbiome of Wild Grapevines Vitis amurensis Rupr. and Vitis coignetiae Pulliat Growing in the Russian Far East"

_plants, 2023, doi:10.3390/plants12162952_

Round 1

Reviewer 1 Report

The ms. “The Endophytic Microbiome of Wild Grapevines Vitis amurensis Rupr. and Vitis coignetiae Pulliat growing in the Russian Far East” (Ms. Ref. No. plants-2542005-v1) presents original results on the assessment of the bacterial and fungal endophytic communities of V. amurensis and V. coignetiae using a genomic approach (through next generation sequencing, NGS) and classical microbiology techniques (endophytes sowing).

Original manuscript: novelty resides in the exhaustive characterization of endophytes from V. amurensis and V. coignetiae.

The investigated topic falls within the aims and scopes of Plants journal and has evident merit.

Major issues:

1.     The ms. is not line numbered, which prevents reviewers from referring to specific lines/paragraphs.

2.     The effect of the endophytic microbiome on the wine making process and on the wine quality should be discussed in the Introduction and emphasized in the Discussion section (see comment #3).

3.     Regarding the fourth paragraph of the Discussion section: In addition to the discussion of the malic acid produced through feermentation with Saccharomyces and the impact on the acidity of wines, it is worth pointing out that high levels of lactic acid were associated with fermentation with the endophytic yeasts, as compared to fermentation with selected strains (please see Hanganu, A.; Todasca, M.-C.; Chira, N.-A.; Rosca, S., “Influence of common and selected yeasts on wine composition studied using 1H-NMR spectroscopy”, Revista de Chimie, 62 (7), 2011, pp. 689-692). This is important to highlight in correlation with the discussion on malic acid, since the malo-lactic fermentation is of outmost importance for the wine quality. Please elaborate on this aspect.

4.     Also, in the same study (Hanganu et al.), it was pointed out that if fermentation is conducted with the spontaneous grapes microbiota generally leads to incomplete fermentation of sugars, thus leading to sweeter wines, as compared to the fermentation with selected strains. This is important especially in the case of artisanal wines (or produced in the household). It is worth discussing this aspect to better point out the applicability of the endophytic strains characterization of your manuscript. Please elaborate.

5.     Regarding the potential use of knowledge on the endophytic microbiota for intrinsic pest protection of vine, it is worth pointing out that phytosanitary protection through chemicals may have a negative impact on the wine quality, especially through production of large of amounts tartaric acid, which can be damaging for the wine because it produces salts with low solubility which can precipitate and thus threaten wine stability (see Todasca, M.-C.; Fotescu, L.; Chira, N.-A.; Deleanu, C. Rosca, S. “Composition Changes in Wines Produced by Different Growing Techniques Examined Through 1H-NMR Spectroscopy”, Revista de Chimie, 62 (2), 2011, pp. 131-134).

6.     The microbial diversity of may also be important for the postharvest quality and decay of fruits, including grapes (please see https://doi.org/10.3390/foods12101980).

7.     Also, complete characterization of endophytes may help in vine adaptation in drought or other stress environment, as demonstrated for other species (please see https://doi.org/10.3390/su13126582, https://doi.org/10.3390/su15043126 and https://doi.org/10.3390/su15021671).

8.     English is generally ok, the ms. can be easily read, however I suggest using the passive voice (as this is the standard for scientific writing). Also, please carefully check the ms. for typos. Please revise.

9.     There are several references missing page range/article number. Please update.

English is generally ok, the ms. can be easily read, however I suggest using the passive voice (as this is the standard for scientific writing). Also, please carefully check the ms. for typos.

Author Response

August 10th, 2023

Dear Editor,

Please find enclosed the revised manuscript (Ms) “The Endophytic Microbiome of Wild Grapevines Vitis amurensis Rupr. and Vitis coignetiae Pulliat growing in the Russian Far East” (plants-2542005). We would like to thank the Reviewers for their time and comments on our work. We carefully examined the Ms according to the Reviewers’ remarks and addressed their comments in our revised Ms using track changes in Word with a brief point by point response to each comment provided separately. We hope that the revised version of the Ms will be suitable for publication in Plants.

Thank you very much for considering our article and we are looking forward to your response.

Sincerely,

Olga A. Aleynova.

Review Report Form Reviewer #1

  1. The ms. is not line numbered, which prevents reviewers from referring to specific lines/paragraphs.

- Answer: We apologize for the lack of lines in the manuscript (MS) and the inconvenience of reviewing the MS. We eliminated this shortcoming and added the MS line numbers.

  1. The effect of the endophytic microbiome on the wine making process and on the wine quality should be discussed in the Introduction and emphasized in the Discussion section (see comment #3).

- Answer: We are grateful to the Reviewer for the high appreciation of our MS. We followed the Reviewer's advice and improved Introduction and Discussion. All changes are highlighted in yellow.

  1. Regarding the fourth paragraph of the Discussion section: In addition to the discussion of the malic acid produced through feermentation with Saccharomyces and the impact on the acidity of wines, it is worth pointing out that high levels of lactic acid were associated with fermentation with the endophytic yeasts, as compared to fermentation with selected strains (please see Hanganu, A.; Todasca, M.-C.; Chira, N.-A.; Rosca, S., “Influence of common and selected yeasts on wine composition studied using 1H-NMR spectroscopy”, Revista de Chimie, 62 (7), 2011, pp. 689-692). This is important to highlight in correlation with the discussion on malic acid, since the malo-lactic fermentation is of outmost importance for the wine quality. Please elaborate on this aspect.

- Answer: Thank you for your comment. We corrected the MS according to the Reviewer's advice: «In addition, application of endophytic yeast leads to an increase in lactic acid content compared to selected yeast strains (Hanganu et al. 2015). It is known that the malolactic fermentation is of outmost importance for the wine quality»

Please see the lines 529-534.

  1. Also, in the same study (Hanganu et al.), it was pointed out that if fermentation is conducted with the spontaneous grapes microbiota generally leads to incomplete fermentation of sugars, thus leading to sweeter wines, as compared to the fermentation with selected strains. This is important especially in the case of artisanal wines (or produced in the household). It is worth discussing this aspect to better point out the applicability of the endophytic strains characterization of your manuscript. Please elaborate.

- Answer: We are grateful to the Reviewer for the valuable comment. We have added information about the fermentation features of endophytic yeast to the Discussion: «Also, the use of spontaneous grape microbiota leads to incomplete fermentation of sugars, thus leading to sweeter wines, as compared to the fermentation with selected strains (Hanganu et al. 2015). This is important especially in the case of artisanal wines (or produced in the household). Probably, the endophytic yeast of wild V. amurensis will be of interest to winemakers to preserve wine acidity or to produce sweeter wines when applying selected microorganisms during the alcoholic fermentation, which can be a promising solution to preserve wine quality and to create wines with new taste qualities»

Please see the lines 531-538.

  1. Regarding the potential use of knowledge on the endophytic microbiota for intrinsic pest protection of vine, it is worth pointing out that phytosanitary protection through chemicals may have a negative impact on the wine quality, especially through production of large of amounts tartaric acid, which can be damaging for the wine because it produces salts with low solubility which can precipitate and thus threaten wine stability (see Todasca, M.-C.; Fotescu, L.; Chira, N.-A.; Deleanu, C. Rosca, S. “Composition Changes in Wines Produced by Different Growing Techniques Examined Through 1H-NMR Spectroscopy”, Revista de Chimie, 62 (2), 2011, pp. 131-134).

- Answer: We have added information about the negative impact of chemicals on the quality of wine in the Introduction: Please see the lines 77-81.

  1. The microbial diversity of may also be important for the postharvest quality and decay of fruits, including grapes (please see https://doi.org/10.3390/foods12101980).

- Answer: We have added information about role of the microbial diversity on postharvest quality of fruits in the Introduction. Please see the lines 51-52.

  1. Also, complete characterization of endophytes may help in vine adaptation in drought or other stress environment, as demonstrated for other species (please see https://doi.org/10.3390/su13126582, https://doi.org/10.3390/su15043126 and https://doi.org/10.3390/su15021671).

- Answer: We have added information about the participation of endophytic microorganisms in plant resistance to abiotic stresses. Please see the lines 436-438.

  1. English is generally ok, the ms. can be easily read, however I suggest using the passive voice (as this is the standard for scientific writing). Also, please carefully check the ms. for typos. Please revise.

- Answer: We have made the corrections and proofread the MS according to the recommendation using passive voice. We also checked MS for typos. All changes are visible in the MS word tracking change mode.

  1. There are several references missing page range/article number. Please update.

- Answer: We have rechecked the references. Thank you very much for considering our article.

Reviewer 2 Report

The authors present analyses of the microbiomes of two wild type non-cultivated Vitis species and they compare them with cultivated grape vines. Their results are very interesting and may contribute to a better understanding on the role of microbiomes in plants regarding plant growth promotion, stress tolerance and pest resistance.

However some issues should be improved in the introduction and discussion sections:

Introduction:

Beginning of the second paragraph: This sentence should provide more information about which plants were used in the study. The reference is also broad, since it is a book, so it is difficult to know the systems the authors refer to

End of that same paragraph: It is also true that endophytes isolated from non-related plants can perform too, plant growth promoting activities, promote stress tolerance and pathogen resistance. This should  be mentioned in the introduction.

For examples see Sanchez-Cruz et al., Isolation and characterization of endophytes from nodules of Mimosa pudica with biotechnological potential. Microbiological Research 218 76–86.

Vazquez-Tapia. et al., Isolation and characterization of psychrophilic and psychrotolerant plant growth promoting microorganisms from a high-altitude volcano crater in Mexico. Microbial Research 232 (126394). https://doi.org/10.1016/j.micres.2019.126394. ISSN 0944-5013

Discussion: 

Page 7 please provide reference(s). Discuss more profoundly why may this happen. It has been proposed that addition of chemical fertilizers and pesticides reduce the amount and diversity of arbuscular Mycorhizae in plants, for example (Sheldrake M., Entangled Life 2022)

Page 7, bottomOomycetes can be serious plant pathogens (Phytophtora and Phytium genera, for example). So this should be taken with care. They may be non-pathogenic in the wild plants but cause desease in the human-cultivated varieties

In general the Discussion can be improved by highlighting, issues such as why in the far away growing vines the diversity is greater; why do the authors think is the reason for different  preferences of different endophytes among the studied species, etc.

Did the authors found microbes which are also in the human gut microbiome? this could add value to their study proposing a source of probiotics when eating certain kind of fruits (grapes in this case)

Author Response

August 10th, 2023

Dear Editor,

Please find enclosed the revised manuscript (Ms) “The Endophytic Microbiome of Wild Grapevines Vitis amurensis Rupr. and Vitis coignetiae Pulliat growing in the Russian Far East” (plants-2542005). We would like to thank the Reviewers for their time and comments on our work. We carefully examined the Ms according to the Reviewers’ remarks and addressed their comments in our revised Ms using track changes in Word with a brief point by point response to each comment provided separately. We hope that the revised version of the Ms will be suitable for publication in Plants.

Thank you very much for considering our article and we are looking forward to your response.

Sincerely,

Olga A. Aleynova.

Review Report Form Reviewer #2

  1. Beginning of the second paragraph: This sentence should provide more information about which plants were used in the study. The reference is also broad, since it is a book, so it is difficult to know the systems the authors refer to

- Answer: We are grateful to the Reviewer for the high appreciation of our manuscript (MS). We have added information about the plant families that were used in the experiments. Also, we detailed the references. All changes are highlighted in green.

Please see the lines 54-57.

  1. End of that same paragraph: It is also true that endophytes isolated from non-related plants can perform too, plant growth promoting activities, promote stress tolerance and pathogen resistance. This shouldbe mentioned in the introduction.

For example, see Sanchez-Cruz et al., Isolation and characterization of endophytes from nodules of Mimosa pudica with biotechnological potential. Microbiological Research 218 76–86. Vazquez-Tapia. et al., Isolation and characterization of psychrophilic and psychrotolerant plant growth promoting microorganisms from a high-altitude volcano crater in Mexico. Microbial Research 232 (126394). https://doi.org/10.1016/j.micres.2019.126394. ISSN 0944-5013

- Answer: Thanks to the reviewer for the valuable comment. We have added information about plant growth promoting activities of endophytes isolated from non-related plants.

Please see the lines 70-75. 

We did not refer to the work of Vazquez-Tapia. et al. 2020, since the study did not use endophytic microorganisms, but yeast and bacteria isolated from rhizospheric soil collected from the Xinantécatl volcano.

  1. Page 7 please provide reference(s). Discuss more profoundly why may this happen. It has been proposed that addition of chemical fertilizers and pesticides reduce the amount and diversity of arbuscular Mycorhizae in plants, for example (Sheldrake M., Entangled Life 2022)

 - Answer: We have added information about the effect of chemical fertilizers and pesticides on the amount and diversity of arbuscular mycorhizae in plants.

Please see the lines 503-507.

  1. Page 7, bottom: Oomycetes can be serious plant pathogens (Phytophtoraand Phytium genera, for example). So this should be taken with care. They may be non-pathogenic in the wild plants but cause desease in the human-cultivated varieties

 - Answer: We added a discussion that concerns Oomycetes.

 Please see the lines 513-522.

  1. In general the Discussion can be improved by highlighting, issues such as why in the far away growing vines the diversity is greater; why do the authors think is the reason for different preferences of different endophytes among the studied species, etc.

-Answer: We added a discussion about why in the far away growing vines the diversity is greater and the reason for different preferences of different endophytes among the studied species

Please see the lines 489-500.

  1. Did the authors found microbes which are also in the human gut microbiome? this could add value to their study proposing a source of probiotics when eating certain kind of fruits (grapes in this case)

- Answer: We have added information about endophytic grape microorganisms, which may be present in the human microbiome and may act as probiotics.

Please see the lines 539-552.

Thank you very much for considering our article.

Round 2

Reviewer 1 Report

The revised ms. (Revision #1) “The Endophytic Microbiome of Wild Grapevines Vitis amurensis Rupr. and Vitis coignetiae Pulliat growing in the Russian Far East” (Ms. Ref. No. plants-2542005-v2) was considerably improved, as compared to its initial form. The authors have carefully addressed all the reviewers’ comments. I do not have further remarks.

Consequently, in my opinion, the ms. “The Endophytic Microbiome of Wild Grapevines Vitis amurensis Rupr. and Vitis coignetiae Pulliat growing in the Russian Far East” (Ms. Ref. No. plants-2542005-v2) now meets the quality requirements for publication in Plants.

Minor English editing tom polish the ms. (however, it can also go in the current form).